# Strategic AI Sabotage: State Attacks on Advanced Systems' Development

## Abstract

Much attention has been given to the possibility that states will attempt to steal the model weights of advanced AI systems. We argue that in most situations, it is more likely that a state will attempt to sabotage the training of the models underpinning these systems. We present a threat modelling framework for sabotage of AI training, including both the necessary technical background and a taxonomy of strategic considerations and attack vectors. We then use this to examine different attacks and assess both their technical plausibility and the mitigations required to defend against them.

## 1 Introduction

As AI systems grow in strategic importance, state-level actors seeking to degrade or corrupt the capabilities of their adversaries may choose to attack these systems through different means. This has received a large amount of attention in the last year or so, with J-AISI recently releasing an illustrative overview of the different ways that AI systems could be attacked (Kiribuchi et al., 2025) and NIST producing an extensive taxonomy of attacks and mitigations relevant to ML systems (Vassilev et al., 2025), among others. Both of these reports—whilst not explicitly calling threats to model integrity 'sabotage'—highlight various different attacks on the training process and make important technical contributions as to those attacks' classification, including within the more general cybersecurity domain.[1] However, they exclusively consider direct, technical vectors and do not go into great detail—inevitably, since the scope of these reports is so wide—and also do not place these attacks within a wider strategic context. More narrowly-focused research has mostly focused on attacks on deployed systems,[2] and the more detailed security work examining the creation of models has not treated the sabotage threat model as a cohesive whole but instead has focused on the customization and fine-tuning of base models[3] or the specific threat of data poisoning.[4]

This paper takes a deeper look at the sabotage threat model, broadly construed, as it is relevant to the initial creation of the models underpinning advanced AI systems.[5] It makes three key contributions:

---

[1] A more detailed placing of *this* work within the existing cyber- and AI-security literature is omitted for brevity, but is substantively the same as can be found in Section 2 of Kiribuchi et al. (2025), which includes a technical classification of these attacks within the taxonomy of the MITRE Atlas—indeed, this paper can itself be seen as a direct follow-up to their recent work, being a combination of a more detailed consideration of some of the attacks they discuss as **C: Model Poisoning**, **D: Data Poisoning**, **J: Adversarial Fine-tuning**, and threat modelling of indirect attacks on organisations. A more detailed comparison of the contributions of this paper compared to the J-AISI and NIST work is attached as Appendix B.

[2] For example, in *Securing AI Model Weights* (Nevo et al., 2024), RAND produced perhaps the most in-depth and comprehensive analysis relevant to securing AI systems but explicitly placed ensuring the integrity of training data and model internals outside the scope of the report.

[3] *See, e.g.* Wan et al. (2023), or indeed the wide variety of work on emergent misalignment following its discovery by Betley et al. (2025).

[4] Indeed, there has been a great deal of investigation into this specific threat area—far too much to cover here, but reading the relevant parts of the NIST report as well as Zhao et al. (2025) will provide a good technical understanding of the current literature; readers should however bear in mind that some data poisoning threat models do rely on somewhat unrealistic levels of access or are not robust to standard levels of monitoring or responsiveness by defenders.

[5] We also touch on, but treat as largely out of scope, actions more drastic than sabotage (e.g. overt military action), attacks which occur after the initial creation of a model (i.e. against deployed systems or in fine-tuning customer-specific models) and specific *consequences* of deploying a corrupted AI model.

- We place the sabotage of advanced AI capabilities in historical context and argue that in several likely near-future scenarios states have the incentive and capacity to use offensive cyber-capabilities to attack the training process.

- We introduce a threat modelling framework in several parts. We provide a technical introduction to the attack surface, then a taxonomy of the different strategic objectives of sabotage, and finally a classification of different attack vectors, examining both direct technical attacks on data ingestion and model training and indirect attacks targeting organisational capacity.

- We use this framework to conduct a threat modelling exercise across the full training pipeline and identify likely attack vectors relevant to the sabotage of AI training. We assess the plausibility of current and near-future exploitation by different threat actors and highlight possible mitigations to prevent this.

## 2 HISTORICAL CONTEXT

In 2009, the Iranian government was advancing its controversial nuclear program amid growing sanctions and pressure. Global opinion—and the pressure of a US populace wary of getting involved in another war—had so far prevented direct military action by Iran's adversaries. However, their nuclear facilities began to experience a high level of unexplained operational failures. Intelligence reports later revealed that Israel and the United States had secretly developed something unprecedented in modern warfare: Stuxnet, the world's first cyber-weapon designed to cause physical damage, which applied irregular and damaging acceleration to centrifuges whilst presenting falsified data to the equipment monitoring them. This sophisticated malware delayed Iran's nuclear ambitions without the risks of overt action.[6]

Historically, states have chosen to sabotage their rivals when facing a significant threat to their security in the following circumstances:

- Diplomatic or economic leverage has proved insufficient, and

- the situation does not yet justify overt military action,[7] and

- when discovery and attribution would fall short of provoking a direct military response or present enough ambiguity to constrain the target from escalating substantially,[8] and

- when the benefit to such an operation outweighs the potential risk to resources and trust from other actors if it were to be discovered.[9]

During the Cold War, both sides conducted sabotage and assassination operations at a level below that which would provoke outright conflict.[10] Stuxnet itself was part of an extended campaign

---

[6]*See* Zetter (2014) for a detailed writeup of Stuxnet, its technical details and geopolitical context. However, Slayton (2016) makes a convincing argument that, ultimately, the delay to the program was only a few months; whilst Stuxnet gives us an excellent illustration of states attempting to achieve their geopolitical aims via cyber-enabled sabotage, it perhaps does not give us an *effective* one.

[7]Whether it be due to lack of or comparative imbalance in military strength (*e.g.* the 1979 Israeli sabotage of nuclear equipment being sold to Iraq by the French), geopolitical constraints (*e.g.* the 1985 bombing of a Greenpeace vessel by the French secret service), domestic politics (*e.g.* the 1954 Guatemalan coup d'etat (Holland, 2005) or the Bay of Pigs invasion), or concerns over the risk of escalation (*e.g.* US support for the mujahideen in the 1980s).

[8]Indeed in the context in the context of the United States, 'covert action' is defined by legislation such that "the role of the United States Government will not be apparent or acknowledged publicly"; *see* Rosenbach & Peritz (2009) for a discussion of this and when such covert action is permitted.

[9]For a detailed explanation of how the value associated with successful sabotage may be orders of magnitude greater than the amount spent by either side and so sabotage can be favoured even in defense-dominant scenarios, *see* Slayton (2016).

[10]For an overview of KGB doctrine and planning around sabotage, *see* Richterova (2024), although it is worth noting (and comparing to recent decades) that actual physical destruction of infrastructure was extremely limited—likely because of escalation risk, and held in contingency for an actual hot war situation—and perhaps the most impactful sabotage was indirect, involving the feeding of faulty schematics and blueprints to the Soviets in the 1980s (Weiss, 1996).

by Israel to sabotage the nuclear ambitions of its adversaries,[11] including assassinations, planting of explosives in third countries, and other sub-military actions. More recently, in 2022, 3 of the 4 Nordstream pipes—taking Russian gas to Europe, its most significant export market[12] and an important source of revenue for the Russian government—were blown up without any conclusive evidence as to who had done it.[13]

In the decades following the Cold War, the traditional boundaries between peace and war have become increasingly blurred as state and non-state actors[14] embrace so-called "grey-zone warfare": a spectrum of hostile activities that fall deliberately below the threshold of conventional armed conflict. This approach allows nations to pursue strategic objectives through a mix of cyber attacks, economic coercion, disinformation campaigns, proxy forces and anonymous military actions, and political interference while maintaining plausible (or at least some) deniability and avoiding the consequences of instigating open warfare. Russia's annexation of Crimea in 2014 created popular awareness of this strategy as it used unmarked and unacknowledged troops, social media manipulation and economic pressure to take control of Ukrainian territory without ever formally declaring war against the country. Similarly, China's so-called "salami-slicing" actions, including the construction of artificial islands and military bases in the South China Sea, the harassment of civilian and military vessels from the Philippines, Vietnam, and other countries, and the reckless use of military assets[15] challenge international law and convention without quite crossing the threshold that would trigger a military response from the United States and its allies.[16]

Traditional executive processes and legal frameworks are currently poorly equipped to respond to threats that do not quite fit into traditional domains of warfare, while the ever-increasing digital transformation of society has created new vulnerabilities to cyber attacks and informational warfare. The last 3 years of war with Ukraine has been accompanied by a corresponding aggressive and extensive campaign of sabotage[17] by Russia against the West—to which it has failed to develop an effective counter (Jones, 2025)—combining a persistent offensive cyber-strategy with more direct bombing and arson of critical infrastructure[18] and attempts to assassinate executives of western companies.[19]

It may be inferred that, for some actors, sabotage would be seen as an attractive option to disrupt strategically relevant technological advancements. Given its strategic importance and potential dual-use applications, it is natural to consider how this context might interact with the rapid increase of investment in, and development of, advanced capabilities in artificial intelligence. The next two sections explore current academic thinking on this and relevant potential near-future scenarios.

## 2.1 MAIM

In *Superintelligence Strategy*, Hendrycks et al. (2025) introduced a three-part plan to manage the development of artificial superintelligence (ASI) covering deterrence, nonproliferation and competitiveness. A key contribution was the introduction of the concept of Mutual Assured AI Malfunc-

---

[11]Known as the Begin doctrine—*see* Talbot (2023) for historical context—although this is only part of their perhaps uniquely effective use of sabotage to degrade the capabilities of their adversaries; *see, e.g.* their recent supply-chain compromise of Hezbollah's pager network (Doran, 2024).

[12]Hooper et al. (2022).

[13]Speculation included Ukraine, Poland, or even the United States, although recent news coverage suggests it was likely Ukrainian operatives.

[14]Or at least, non-state actors with some plausible deniability as to not being state-directed, *see, e.g.* Chinese and Russian commercial shipping vessels repeatedly 'accidentally' cutting undersea infrastructure in the Baltic and around Taiwan (van Soest, 2025; Daud et al., 2024).

[15]The Hainan Island incident is well-known (Donnelly, 2004); *see also, e.g.* a recent incident involving Chinese sonar and Australian divers.

[16]For a more comprehensive detailing of these tactics, *see* Helmus et al. (2024) p. 25 and Chapter 2 more generally; Derek Grossman has also written two relevant opinion pieces (Grossman, 2024b;a).

[17]Note that whilst there is an ongoing military campaign against Ukraine itself, Russia has evidently not decided that the situation justifies overt military action against the targets of their sabotage, i.e. Poland, Czechia, UK, etc., and has chosen to attempt sabotaging their ability to help Ukraine instead.

[18]*See, e.g.*, the growing list of explosions at European munitions factories (**?**) and the recent destruction of a railway link between Poland and Ukraine (**?**).

[19]Covered at some length in this writeup by CNN.

tion, or *MAIM*. With this, they outlined "a deterrence regime resembling nuclear mutual assured destruction [...] where any state's aggressive bid for unilateral AI dominance is met with preventive sabotage by rivals". In practice, this predicts a dynamic where the training of advanced AI systems is prevented by *covert sabotage*, *overt cyberattacks*, and (less relevant for us) *direct kinetic action*, resulting in the substantial slowing or stopping of the creation of such systems.

The academic response to this has been mixed; Abecassis (2025), of MIRI, responded, claiming that in practice this dynamic would not apply because of the "lack of effective, monitorable, and clear red lines". RAND also responded, with Rehman et al. (2025) suggesting that the game theory assumptions do not hold in practice and MAIMing actions are highly likely to be escalatory. To determine whether these criticisms are reasonable, and whether we should expect the MAIM dynamic to actually hold, we need to understand how sabotage might occur and know the plausibility and scope of particular attacks, not least because if MAIM is implausible we must seek other ways of ensuring stability.

## 2.2 LIKELIHOOD OF SABOTAGE

Various world leaders and governments have indicated that they see AI as a potential path to strong military and economic advantages.[20] As such, there has been substantial academic attention given to the strategic implications of the development of advanced AI systems in the domain of international relations, geopolitics and the military balance of power.[21] Correspondingly, there has been substantive discourse around the possibility that adversaries will attempt to *steal* the model weights underpinning these systems and thus reduce the relative military or economic advantage gained by the actor which has developed advanced AI capabilities.[22] We consider this to be a likely course of action in some circumstances—specifically, when one party leads in some area of AI research, but another is capable of 'fast-following', has the capacity to benefit substantially from the deployment of the stolen models, and has viable pathways to obtain them without excessive risk or resource expenditure—but there are several more likely scenarios where we expect a state actor[23] to prioritise sabotage over theft of AI systems instead (for a more general look at the different forms of sabotage and their strategic implications, see Section 3.2). We argue that these scenarios include cases when the actor:

- does not have the compute to deploy advanced models for strategic purposes, or alternatively the economic integration, and believes it is imperative to stop rivals who would have this capacity.[24]

- has their own, better models, which they believe to be secure and want to lower the possibility of rivals catching up.

- believes that their rival could use the model being developed to attain a decisive strategic advantage or otherwise take significantly destabilising actions.[25]

- wishes to subvert the model before it is deployed, causing it to malfunction or underperform in specific circumstances.

---

[20]For example, Putin, Biden, Trump, Vance, Xi, among others.

[21]*See, e.g.*, Pavel et al. (2023), Scharre (2023), as well as the recent UK Strategic Defence Review (UK Ministry of Defence, 2025), which highlighted several different candidates for near-future military use of AI capabilities.

[22]For a comprehensive understanding of this, *see* RAND's seminal report *Securing AI Model Weights* (Nevo et al., 2024).

[23]Of course, there are also *commercial* reasons that companies might sabotage their rivals, which we do not consider.

[24]After all, many of the states commonly acknowledged to invest the most heavily in offensive cyber capabilities—including Russia, North Korea, Iran, and to a lesser extent the United Kingdom and Israel—are not engaged in a compute buildout comparable with the United States or China (International Institute for Strategic Studies, 2021; Pilz et al., 2025).

[25]Mitre & Predd (2025) provides an in-depth analysis of this scenario, among others relevant to the balance of power.

- similarly, wishes to instil secret loyalties to a foreign state—or even a specific individual, which might be harder to detect—into a model, in preparation for a subsequent military crisis or attempt to perform a coup.[26]

Additionally, a state actor might *prefer* to steal the model weights if possible, but finds it infeasible to gain access to lab systems—for instance, if the lab has achieved SL4[27] or higher—or similarly, has full access now but expect that they will lose it at some point because the lab is investing heavily in security improvements, giving any capacity to interfere with ongoing training a use-it-or-lose-it element. In these circumstances, sabotage may be seen as an acceptable fall-back option; in general we expect that sabotage would require a lower level of access because the attack surface is much larger—a lab must defend not only their training and deployment systems but also all the upstream inputs to model creation, including both human and technological capital.

Current AI capabilities already provide significant uplift to attackers at multiple stages of the cyber kill chain and it seems inevitable that this will increase in the near future.[28] Multiple frontier labs have documented how state-sponsored and organised criminal groups are using their models to accelerate and enhance their capacity to attack targets.[29] Without concerted efforts by model developers to reach the frontier of cyber-defense they will become more vulnerable to breach by state-sponsored groups, failing to take advantage of AI-enhanced defensive possibilities; these attacks themselves will become harder to attribute.[30] For actors worried about the risk of escalation, this shifts the calculus towards sabotage rather than 'merely' model theft.

## 3 A TAXONOMY OF SABOTAGE

### 3.1 TRAINING OF ADVANCED AI SYSTEMS

The training of an advanced AI system can be split into different stages.[31] Understanding the broad technical motivation and implementation of each of these is essential for effective threat modelling.[32]

The first stage is **design and planning**. This includes choosing the model architecture (transformer, mixture of experts, etc.), deciding on model size and other key training and design parameters. It also includes operational planning, which might include organisational and budgetary considerations, testing and evaluation schedules, and the design of the training pipeline itself. Concurrently, it requires the acquisition and build-out of large amounts of compute capacity, critically including the GPUs, storage and networking hardware needed for training modern AI systems.

The second stage is **data gathering and preprocessing**. Companies must gather large amounts of data across the domains they wish their model to be capable of. This can be sourced from publicly available sources—for example, by crawling the internet to obtain general text content or scientific papers—or the model developer may pay for access to non-public material (books, stock images, music) or for expert generation of relevant training data. This data must be high-quality, so

---

[26]*See* Davidson et al. (2025) for a detailed examination of this threat model.

[27]Nevo et al. (2024) defined five **S**ecurity **L**evels, of which the most relevant to us are **SL4** ('A system that can likely thwart most standard operations by leading cyber-capable institutions') and **SL5** ('...could plausibly be claimed to thwart most top-priority operations by the top cyber-capable institutions').

[28]Unit 42, Palo Alto Networks (2025); *see also* this thread on X by Dawn Song.

[29]Anthropic (2025a) details use of their models to conduct a data theft and extortion campaign; Google Threat Intelligence Group (2025) provides an extensive analysis of the variety of cyber-offensive capabilities used by different state-sponsored groups.

[30]*See* Murphy & Stone (2025) p. 10 for discussion of attributability and pp. 11–13 for technical analysis of how for most organisations there is differential uplift favouring attackers.

[31]In practice these first two stages may overlap substantially—data acquisition often begins before architectural decisions are finalized, and indeed the availability of data may well influence the chosen model size. We present them as distinct stages here so as to better elucidate the different attack surfaces and threat vectors relevant to each.

[32]Note that while many well-known AI systems today build upon the "transformer architecture", this description is slightly more general and so our threat modelling may also be applicable to the training of other advanced AI systems. However, by necessity this overview contains significant simplifications, and threat modelling in a particular context should account for system-specific technical details, upstream dependencies, and organisational factors.

preprocessing is then performed to remove duplicate, low-quality or inappropriate content and fix formatting issues.[33]

The third stage is **pretraining**. A model is fed the vast amounts of data gathered in the previous step and learns general capabilities. For LLMs, this involves repeatedly giving it example text, asking it to continue the document, and then altering the model internals to make it more likely to predict the 'correct' answer in future. This step is the most computationally expensive - it is estimated that the pretraining of GPT-4 cost over $100 million[34] and used over 60 million GPU hours to process around 10 trillion words of content.

The fourth stage is **post-training**. The aim of this is to shape how those capabilities developed in the previous step are expressed and ensure that the model behaves appropriately in actual real-world applications. Initially this will be via supervised fine-tuning—curating high-quality examples of desired behaviour, such as answering questions, following instructions, or behaving as a chatbot, and then updating the model to produce outputs more similar to this. This is typically followed by reinforcement learning techniques like RLHF,[35] where the company gets internal or external annotators to compare possible model outputs and rank them according to how much they align to a chosen set of values, such as being Helpful, Harmless and Honest (HHH). More general safety and capability testing is also performed at this stage.

## 3.2 STRATEGIC CONSIDERATIONS

Sophisticated threat actors may attempt to sabotage AI training runs with two primary strategic objectives, each serving distinct operational goals and requiring different defensive approaches. **Capability degradation** attacks aim to reduce the performance and reliability of AI systems at specific tasks or in general capabilities, without necessarily changing their high level objectives. By contrast, **value misalignment** attacks aim to redirect the high-level goals of AI systems towards attacker-preferred objectives without necessarily reducing its capability to perform any particular task.

Sabotage may be meaningfully classified further on several axes: it may be **overt**, where the attacker does not care or intends that the developers notice the model does not meet their expectations (and possibly that sabotage has been attempted), or **covert**, where the attacker aims for the developer to deploy the AI system without noticing it is corrupted. It can also be **attributable**, where the model developer (perhaps with the aid of state counter-intelligence) could feasibly assign blame to a specific threat actor, or **anonymous**, where this is not realistically possible. Sabotage implementation can also be split based on technical and implementation details; this is continued in Section 3.3.

The choice of a particular method of sabotage will depend on the strategic objectives and technical capacity of the specific threat actor. The implications and examples of these choices can be found in Table 1.

## 3.3 ATTACK VECTORS

Each of the sabotage objectives outlined in the previous section can be achieved through various technical and non-technical attacks by different actors which we outline these in this section.[36]

We can split the attack vectors relevant to sabotage of model development into **data poisoning**, **model poisoning**, and **process disruption**.[37] Broadly speaking:

- **Data poisoning** - inserting malicious, adversarial examples into the dataset used for training the system. This may be broad (generally degrading capabilities, implanting general

---

[33]Lee et al. (2021).

[34]Knight (2023).

[35]**R**einforcement **L**earning from **H**uman **F**eedback, though this is being replaced by more scalable methods such as **R**einforcement **L**earning from **AI F**eedback (RLAIF), *see* Lee et al. (2024).

[36]Note that we do not consider attack vectors which do not affect the development of the AI model itself, such as 'jailbreaking' or prompt injection, extraction of sensitive training data, adversarial post-deployment finetuning, or any inference-time attacks.

[37]See the J-AISI report (Kiribuchi et al., 2025) for a more general look at attacks on AI systems, although it does not cover 'process disruption' or supply-chain attacks.

Table 1: AI sabotage strategic impacts.

| Type | Impact | Strategic value | Attack vectors |
| --- | --- | --- | --- |
| DO | Models fail benchmarks, show reduced accuracy or do not meet expectations. Development delays, customer disappointment. Lower researcher morale. | Slow rival timelines. Force expensive validation cycles. Fragmentation of research efforts. Suspicion and paranoia of insiders. | Data poisoning, cyberattacks, social engineering, personnel targeting, political campaigns |
| DC | Specific capabilities degraded without detection. New research directions are poorly guided. Mediocre progress. | Comparative advantage for frontier models reduced, research efficiency impaired, investor confidence in AI companies eroded. | Amplifying conflicts, honeypotting and coercing researchers, data poisoning |
| MO | Models exhibit obvious undesirable behaviours, violate safety guidelines. Negative press if released anyway. | Damage to reputation and public trust, especially for safety-critical applications. Internal discord. | Alignment-targeted poisoning, exploiting emergent misalignment |
| MC | Model passes evaluations but contains hidden malicious capabilities activated by triggers post-deployment. | Enable military/espionage exploitation through hidden backdoors. | Insider corruption, blackmail, deep cover placement, adversarial fine-tuning |

Attack type = **D**egradation vs. **M**isalignment + **O**vert vs. **C**overt

ideological or philosophical values) or narrow (reducing performance on a specific task or in a given domain, or planting a backdoor where the model malfunctions in specific circumstances).

- **Model poisoning** - altering the creation of the model itself in ways other than modifying the training data. This might include altering the training parameters or modifying the model internals directly during the training process.

- **Process disruption** - slowing or stopping key parts of the development process. In the absence of other forms of sabotage, this does not necessarily affect the final product but may make the training more expensive, less efficient, and delayed.

Each of these vectors may be via **direct** attack on the training process (corrupting upstream data, poisoning internal sources of information, cutting the power to a data centre, etc.) or by **indirect** attack on the organisation's capacity to manage the training of AI systems (via political means, uncovering scandals within the organisation, assassinating key researchers, etc.). They can be **internal**, where the attack requires access to the systems of the lab developing the system, or **external**, where the target is an upstream input of the training process.

The efficacy of these attack vectors will vary significantly depending on model architecture and organisational structure and so the required mitigations will change correspondingly. For example, data poisoning attacks are substantially easier to execute in federated or multi-organization collaborative training—where each participant contributes data with limited oversight—than in centralized systems with comprehensive monitoring.[38] Similarly, continuous online learning systems can be highly vulnerable to gradual poisoning that would be readily detected in discrete training runs with clear before/after comparisons but evades scrutiny when spread over many incremental updates. Cloud-based training may offer superior monitoring and anomaly detection through mature security operations, but expands the attack surface to include cloud provider personnel and increases system

---

[38]Bagdasaryan et al. (2019) demonstrate how a single attacker can have a drastic impact in a federated setup; *see also* Tolpegin et al. (2020) and Sun et al. (2020).

complexity that can obscure attribution of disruptions. These architectural considerations inform our assessment of specific threats in Section 4.

## 4 THREAT MODELLING

We used the framework outlined above to conduct a threat modelling exercise considering sabotage at each point of the training pipeline. Whilst inevitably limited by lack of access to non-public information regarding lab security practices, adversary capabilities, and model architectures,[39] we were particularly concerned by some threats and present a selection of them below, along with an assessment of their plausibility and possible mitigations.[40] To aid those who wish to use our framework for a similar exercise on their own systems, we have created a 'checklist' of questions relevant to the attack surfaces, strategic context, and defensive possibilities of training a specific system; this is attached as Appendix A.

**Model poisoning and research disruption by trusted insiders.** A researcher or other employee with access to internal systems has been employed by foreign intelligence since before they joined the company. Alternatively, they have become disillusioned with or ideologically opposed to actions taken by their employer. Alternatively, they have been bribed, blackmailed, or otherwise coerced. They may disrupt experiments or manipulate results, including those conducted by other researchers. If sufficiently senior, they may advocate for suboptimal or ineffective research directions.

We consider this reasonably likely, although the efficacy of disruption is uncertain and this level of access may be preferentially used for stealing model weights instead, if strategically favoured. Companies being compromised by intelligence agencies is not a new threat, though known cases were more often aimed at exfiltrating information rather than directly sabotaging research.[41] There was extensive coverage around allegations that a single intern at Bytedance had caused a large amount of disruption in an attempt to sabotage rival projects, including successfully evading counter-measures and investigation for some time,[42] which suggests that disrupting research in a deniable fashion maybe not be *that* hard, especially for a researcher with more experience or outside technical assistance. However, there will be a trade-off between effectiveness and evading detection (although in practice normal experimental variance may make disruptions of a few percent virtually undetectable). Modern training runs are well monitored and designed to be resilient to disruption, so it would likely be more effective to use such a researcher to surreptitiously subvert security measures in a deniable fashion and rely on subsequent external cyberattacks to cause disruption. As an alternative, attackers may find corrupting the evaluation process to be be higher value, causing labs to miss safety concerns, underestimate and subsequently release dangerous capabilities[43] or fail to detect other sabotage. Depending on lab security posture, this may be significantly more feasible with less control on operations, more people involved and higher stochasticity in results than model training itself.

Stronger, and repeated background checks (as recommended in Nevo et al. (2024)) would go some way towards mitigating the risk here but would have other costs to research velocity. Promising technical mitigations might include two-person integrity for control of critical experiments, as well as mandatory code review by senior researchers and AI-driven anomaly detection in user file- and

---

[39] Indeed, this approach is limited by its generality and both attacks and mitigations discussed here may not apply to all or even most training processes. We recommend security professionals considering the sabotage threat model for their specific context use our framework as a high-level tool to identify areas of concern and then consider specific threats within each area by using a standard industry model such as STRIDE as appropriate.

[40] To minimise the risk of our work aiding threat actors we omit, for now, detailed analysis of threat pathways and do not include attack vectors new to the literature. We are seeking review and guidance from relevant experts and stakeholders on presenting our research publicly and responsibly.

[41] The CIA secretly controlled a Swiss cryptography company—Miller (2020) gives an in-depth exposition, and is a worthwhile read—for decades, Twitter was compromised by Saudi Arabia to deanonymise dissidents (United States v. Abouammo, Complaint, No. 3:19-71824 (N.D. Cal. 2019)), and North Korea has an ongoing campaign to place insiders in American tech companies (FBI, 2025).

[42] Reuters (2024).

[43] This threat model is explored and considered feasible in Benton et al. (2024), although in their case the adversary is the misaligned system itself rather than an external actor.

system-access, commits, and other behaviour. Each of these would also reduce the risk of sabotage following theft of user credentials.

**Harassment of key researchers.** The identity of researchers at frontier labs is not currently, typically, secret. States may target key personnel, disrupting their lives substantially, including fraudulently targeting their finances, compromising communication and social media accounts, as well as those of their friends and family,[44] until they stop working on frontier capabilities. Attribution is difficult, especially since these tactics are primarily cyber-based; attackers would plausibly hide behind the guise of an anti-AI 'hacktivist' group. Alternatively, social engineering might be utilised to manipulate key personnel into taking insecure actions, similar to the 2024 compromise of XZ utils,[45] or to cause significant psychological distress.

Our assessment of plausibility depends on the scope; this would be seen as a severely hostile action but it is unclear how anonymous a campaign like this could potentially be. In individual cases, it would be very difficult to distinguish this from more common harassment of public figures. Even moderate harassment can substantially impact wellbeing and cognitive performance.

Defenses against this might include lab support for various forms of security training and protection of researchers as well as regular psychological assessment and support.[46] Involvement of government by treating systemic harassment of researchers as a national security concern may also be an option.

**Adversarial fine-tuning to induce misalignment during post-training.** A compromised insider might corrupt data to place a backdoor during post-training. Alternatively, a third-party data provider is compromised and select RLHF data is poisoned, leading to models developing strongly misaligned values in specific contexts.

It seems unrealistic that all upstream data providers can reach a sufficient level of security to prevent compromise by state actors.[47] However, detecting and preventing data poisoning is an area of significant current research interest. Researchers have demonstrated using weaker LLMs to filter data used for fine-tuning more advanced models,[48] although recent work by Souly et al. (2025) suggests that successful poisoning of larger models (in pre-training) can be achieved with corruption of a proportionally-smaller subset of the training data, making prevention increasingly difficult.

There are a variety of white-box and black-box methods which offer promise in finding backdoors in models,[49] but other work—where backdoors are specifically trained to avoid detection by known techniques—has shown considerable efficacy in evasion[50] and a state-level adversary would presumably be willing to devote substantial resources to research of this. It is not clear to us which side 'wins' in the limit and we would consider it unwise to rely solely on interpretability to rule out model misalignment. Even if a backdoor is detected, the possibility and efficacy of removal is an open question. It *may* be possible to remove backdoors,[51] but the efficacy of current methods may be overstated or misleading.[52] Post-deployment mitigations might also be possible, with model

---

[44]*In extremis* a state might choose to make coordinated attack to disrupt an organisation similar to that deployed against ISIS in Operation Glowing Symphony.

[45]This was a *fascinating* compromise which was both technically skilled and involved a multi-year operation to build the trust of the key maintainer whilst adversely affecting his mental health. *See* Akamai Security Research Team (2024), CSO Online (2024) and this thread on X; it appears that we were collectively very lucky that it was caught when it was.

[46]This would ideally be mandatory to prevent stigma around accessing this, though careful attention will be needed to avoid a similar situation to the FAA, where pilots routinely conceal relevant information due to worries about having their licenses removed (Cross et al., 2024).

[47]They may also be based in or have a substantive presence in a different jurisdiction, and vulnerable to direct coercion by the authorities there.

[48]*See* Li et al. (2024), although we remain concerned about federated or online training, particularly when 'privacy-preservation' mandates that only gradient-updates are shared.

[49]*See, e.g.* Anthropic's work on detecting so-called 'sleeper agents' (MacDiarmid et al., 2024).

[50]Sahabandu et al. (2024).

[51]Possibly even without concrete knowledge that one is even there, *see* Goldwasser et al. (2024).

[52]Zhu et al. (2024).

providers monitoring output for different forms of undesired content,[53] and models being embedded in an architecture which validates its decisions against a core set of human-readable principles.[54]

## 5 Conclusions

We have argued that state-sponsored sabotage of AI training represents a threat worthy of serious consideration by the developers of advanced systems, giving significant historical evidence of states seeking to use sabotage against their rivals and analysing modern AI development and its implications in this context. We have presented several scenarios where sabotage may be strategically favoured over the theft of model weights, including when adversaries lack the capacity to use them effectively, seek to prevent rivals from having access to powerful capabilities, or aim to embed hidden backdoors in systems deployed by their adversaries.

We have presented a threat modelling framework for the sabotage threat model which classifies sabotage objectives and attack vectors, allowing AI developers, security practitioners, and policy-makers to think strategically about sabotage both generally, and in specific defensive contexts. Our threat modelling exercise identified particular areas of concern and possible defensive mitigations, though detailed assessment of attack plausibility, effectiveness, and preventative measures would require access to non-public information regarding AI research, lab security posture and adversary capabilities. The possible attack surface we outline is very large with significant uncertainty about vulnerability levels and the feasibility of comprehensive protection against state actors.

We conclude that urgent work is needed to defend against state sabotage of AI training runs, beginning with a more comprehensive understanding of the different attack and defence possibilities and their relative likelihood.[55] The stakes are high, and we in the security community have a part to play in enabling informed decision-making by both labs and policymakers.

### Ethics Statement

Cybersecurity research is often inherently dual-use. However, our work is, relatively speaking, strongly defensive in nature. We provide a framework to be used by the developers of AI systems to protect the integrity of their models, and make a case for policy-makers to treat a particular threat model seriously and invest in defensive cybersecurity.

Whilst we do highlight—and thus raise awareness of—potential attack vectors, we provide no code or other detailed aid to those aiming to exploit them. At the request of relevant figures within government, detailed analysis of specific, novel attack vectors has been removed from the public submission pending further review.

There are substantial risks associated with deploying a model which has been corrupted.[56] We are confident that working towards preventing this is strongly net-benefit, whilst acknowledging that much work to increase the capabilities of advanced AI systems does bring itself additional risks to those affected by the system.

The MAIM dynamic relies on the effectiveness of sabotage to maintain stability. It is important to understand whether this is actually the case to inform whether we should seek other means of ensuring stability.

---

[53] This is already the case for CBRN uplift, *see* Anthropic (2025b); OpenAI (2023).

[54] Although our threat model here is 'intentional' misalignment rather than 'accidental' or 'naturally occurring', there is substantial overlap with the methods and mitigations studied as part of the so-called 'AI Control Agenda'; *see* Greenblatt et al. (2024) and subsequent work by the UK AI Security Institute (Korbak et al., 2025).

[55] Indeed, an excellent next step would be conducting a more detailed threat modelling exercise analogous to RAND's work on securing model weights (Nevo et al., 2024). Additionally, a game-theoretic formalization of the conditions under which sabotage dominates theft would substantially improve our understanding of the MAIM dynamic and the strategic likelihood of training disruption.

[56] An adversary may have trained specific malicious behaviours as part of their sabotage, or even just disrupted safety and alignment training to produce a model which is generally misaligned.

USE OF LLMS

Claude Sonnet 4 was used extensively for literature review as well as for some assistance with formatting and sentence structure.

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

## A PRACTITIONER CHECKLIST

We present a list of questions regarding attack surfaces, strategic context, and potential mitigations, that security and ML practitioners may wish to consider when creating advanced systems. Questions are split following the delineation outlined in Section 3.1. Note that this list is not exhaustive and part of it may not be applicable, but should provide a good starting point for threat modelling of any particular system.

Questions in **bold** we consider particularly important, questions in *italics* are likely relevant only at high security levels (SL4+).

### A.1 GENERAL

- **Are your logs tamper-resistant and tamper-evident? How is this verified? How often? How comprehensive are they?**

- Do you have forensic capabilities to attribute sabotage (including access logs) and understand its scope? Is your logging sufficient for investigation, and how long is it retained?

- **How do you plan to maintain safety and security even if the model is subtly corrupted—with controls not relying solely on model alignment?** Do you validate outputs against documented and readable principles? Do you have monitoring which could detect unwanted behaviour in deployment?

- What post-deployment monitoring might detect activation of hidden capabilities or backdoors? Do you analyse behavioural patterns across users? What is the process for investigating user reports of strange behaviour?

- **What is your plan for dealing with insider threats?** Do you have background checks, how stringent are they, and how often are they repeated? Is this dependent on threat assessments, seniority, level of access etc.? What indicators might suggest compromise, coercion, or ideological opposition? Do you use technical means to automatically monitor for indicators of compromise (including theft or 'accidental' leak of credentials)? Is there a process for reporting concerns about colleagues?

- **Do you provide mandatory security awareness training? Is it any good (i.e. specific to your context, not just generic 'phishing emails tests' and quizzes about connecting to public WiFi)?** What about psychological support and counselling? Do researchers recognize social engineering? Is support accessible without career consequences, as far as practically achievable?

- How would you distinguish normal variance in evaluation from deliberate low-level sabotage? Do you investigate anomalies or just rerun? Is there sufficient logging for after-the-fact investigation? If you suspected sabotage, what would you actually do? What evidence threshold triggers internal investigation, government/natsec involvement, or public disclosure?

- *Do you have relationships with government or intelligence community for threat intelligence? Are you receiving threat briefings? Do you have a process for reporting suspected state-actor activity?*

- In the process of improving security adversaries with existing access may act before losing it. How will you handle this? Do your improvements include testing for existing compromise to prevent persistence of prior access?

- Do you conduct red-team exercises specifically targeting the sabotage threat model? *Are red teams given realistic state-actor constraints (which may include assuming high levels of knowledge and access to internal systems)? Do you employ external red teams using methods unknown to internal teams?*

### A.2 DESIGN AND PLANNING

- Do you have a formal policy mapping roles to levels of access to systems and data? Is the documentation of architecture, configuration and implementation decisions version-controlled with attribution? What about the technical implementation? Do you have infrastructure as code? Can you reconstruct a full history of modifications?

- *What is your hardware procurement policy regarding country of origin and integrity of firmware? How do you know exactly what firmware is running? Is it cryptographically signed? Do you monitor for anomalous hardware behaviour? If there undetectable compromise is there anything you can do about that?*

- Have you assessed dependencies on third-party cloud providers and their personnel security? What contractual security requirements exist? Where is the hardware physically located? Could political or regulatory pressure in relevant jurisdictions lead to compromise?

- What contingency plans exist if key researchers leave or become otherwise unavailable? What is your "bus factor"? Is institutional knowledge documented or siloed (either at an individual or a team level)? What support exists if personnel become targets of harassment campaigns by state or anti-AI actors?

### A.3  DATA GATHERING AND PREPROCESSING

- What proportion of training data comes from sources outside your direct control, and do you have provenance tracking? Do you sample and manually review external data? Do you timestamp everything to retrospectively handle data poisoned during specific periods?

- Have you mapped the geographic footprint and jurisdiction of data providers? Do any have operations in jurisdictions with mandatory data access laws? Have you tiered sources by trust level?

- What anomaly detection exists for poisoned or adversarial examples? Do you use statistical methods to detect distribution shifts? Can you use weaker models to filter data for stronger models? What's your sampling rate for human review? Is any data obtained where this is impossible (i.e. gradient-only updates) and is it worth the risk?

- What capability-specific evaluations might catch targeted degradation? Are you SOTA with public literature?

- **Who has access to modify preprocessing and filtering pipelines, and are changes version-controlled and peer-reviewed?** Do you (can you) validate filtering outputs by sampling? Could someone introduce rules targeting data relevant to a narrow backdoor? Could they systematically filter *out* data in a way which would corrupt the training?

### A.4  PRE-TRAINING

- Who has authority to modify training hyperparameters, and what approval processes exist? Is there version control and audit logging for all of these? Is there separation between who proposes and who approves changes? **Are two-person integrity controls or similar measures enforced technically enforced through your CI/CD? Who can bypass this?**

- What monitoring exists for unexpected changes in loss curves, gradient statistics, or other training dynamics?

- At what level of capability degradation would you investigate rather than attribute to noise? What's normal variance in your evaluations? If you noticed 2% degradation, what would your investigation process be?

- **How is access to training infrastructure controlled, and would you detect credential theft? Is MFA required?** Do security controls have a 'fail-deadly' setup? Are users locked out after inputting incorrect credentials more than once, or correspondingly all logins after an unusual number of failures (to prevent 'password spray' attacks? Does this itself cause a vulnerability to Denial of Service attacks? Does this matter?

- **How are code commits to training infrastructure reviewed? Are reviewers trained to consider security issues? Do you require review by someone outside the author's direct collaborators? What verifies executed code matches reviewed code? Can code only be deployed by automated processes or is there a 'hack' to get round controls for speed?**

- What integrity verification exists for checkpoints? Are they cryptographically signed and read-only after creation? Is all access and modification (if possible) logged? Do you verify integrity at each step?

- Could an insider cause disruptions that appear accidental? Does your auditing have the specificity to distinguish accidents from sabotage?

## A.5 POST-TRAINING

- **Who is running the evaluations, and who has access to evaluation frameworks and data? Are safety-critical evaluations hosted on separate infrastructure with stricter access controls? Is there external validation? What level of access would it take to alter previous results, and how do you validate they have not been untampered with?**

- How long would it take to notice 1% vs 10% benchmark degradation on any particular benchmark you care about?

- What proportion of preference data comes from external annotators, and how are they vetted? Do you have visibility into individual annotator behaviour? Could a single annotator meaningfully bias preferences? Could coordinated annotators appear independent? **What jurisdiction are they in (and thus could the entire org be compromised)?**

- Do you have quality control that might catch subtly poisoned preference data? Do you test for context-dependent behavioural shifts? For RLAIF, what verifies the feedback AI hasn't been compromised?

- Are you implementing published backdoor detection methods such as probing for sleeper agents? Do interpretability researchers validate deployments? What triggers deeper investigation? What white-box interpretability methods are you using to detect hidden capabilities or other misalignment? Are these teams distinct from those with control over training?

- Do evaluations test for triggered behaviours, context-dependent misalignment, or other 'backdoors'? Do you red-team for these sorts of behaviours? Do you test with trigger patterns relevant to state actors (geopolitical contexts, specific nations, timeframes)?

- Do you conduct formal alignment audits at key checkpoints? Who conducts them? Internal, external, or both? How do you ensure auditors haven't been compromised or left loopholes so a corrupted model could pass? Do you use multiple independent auditors?

## B COMPARATIVE TAXONOMY

In this section we provide a comparative taxonomy with J-AISI (Kiribuchi et al., 2025) and NIST (Vassilev et al., 2025) classifications. Note that the assessment is only in the context of sabotage of the training process itself, not post-deployment attacks,[57] and only covers the classification of attacks rather than any other contributions of these works.

---

[57]For example, NISTAML.027 *does* discuss generating 'content that deviates from benign behavior to align with adversarial objectives' but only in the context of attacking a deployed system.

Table 2: Comparative taxonomy with J-AISI and NIST classifications.

| Element | J-AISI | NIST | Notes |
|---|---|---|---|
| *Strategic Objectives* | | | |
| Degradation | ✓ | ✓ | Both discuss degradation of capabilities in broad and narrow senses (wide vs. targeted poisoning). |
| Misalignment | ✗ | ✗ | Neither considers corruption of alignment—goals, values, or loyalties. |
| Overt / Covert | ○ | ○ | Both address detectability as a *technical property* of attacks, but not as a *strategic choice* by the attacker. |
| Attributable / Anonymous | ✗ | ✗ | No consideration is made to attributability, which is out of scope. |
| *Attack Vectors* | | | |
| Data Poisoning | ✓ | ✓ | Well-covered by both in NIST §2.3.1–2.3.3 (NISTAML.012, 013, 021, 023, 024) and J-AISI Attack D. Data Poisoning and arguably Attack J: Adversarial Fine-tuning. |
| Model Poisoning | ○ | ○ | Our threat model is broader. NIST/J-AISI focus exclusively on technical mechanisms of attack: altering hyperparameters, compromise of the implementation of the training algorithm, etc. (J-AISI C: Model Poisoning; NISTAML.011, 026, 051). We additionally address strategic insider sabotage: research misdirection, disruption of experiments, evaluation corruption, etc. |
| Process Disruption | ✗ | ✗ | Neither considers attacks on the training process itself (researcher targeting, infrastructure attacks) as distinct from attacks on data or models. |
| *Attack Modalities* | | | |
| Direct | ○ | ○ | Both cover direct attacks on training extensively, but our threat model is slightly wider and includes e.g. disruptive cyberattacks during training. |
| Indirect | ✗ | ✗ | Indirect attacks on organisational capacity (political pressure on potential data providers, personnel targeting, amplifying conflicts) are out of scope. |
| Internal / External | ○ | ✓ | NIST has a formal supply-chain category (NISTAML.05); J-AISI mentions it briefly in Attacks C and D. |

✓ = Explicitly covered with comparable scope; ○ = Partially addressed; ✗ = Not covered.

