# OpenReview forum: "Strategic AI Training Sabotage: State Attacks on Advanced Systems' Development"
_ICLR.cc/2026/Conference — Submitted to ICLR 2026_

### Official Review · Reviewer_x7UV · 2025-10-15

**Soundness:** 3
**Presentation:** 4
**Contribution:** 4
**Rating:** 4
**Confidence:** 5

**Summary:**

The paper “Strategic AI Sabotage: State Attacks on Advanced Systems’ Development” examines how state actors could target the training pipelines of advanced AI systems as a form of strategic sabotage. It proposes a threat modeling framework outlining sabotage goals, attack vectors, and motivations, analyzing methods such as data and model poisoning that could degrade performance or delay rival progress. Drawing parallels to Stuxnet and Cold War covert operations, the paper situates AI sabotage within modern grey-zone warfare and calls for stronger defensive measures, cooperation, and safeguards to protect AI development from state-level interference.

**Strengths:**

The paper is highly original, introducing the idea of state-level AI training sabotage as a new and important threat. It effectively blends technical and geopolitical analysis, supported by clear threat modeling, a well-defined taxonomy, and credible historical analogies. The writing is clear and well-structured, with visuals that enhance understanding. Overall, it makes a significant and timely contribution by spotlighting an overlooked risk at the intersection of AI safety, national security, and global governance.

**Weaknesses:**

The paper is largely conceptual, lacking empirical validation or quantitative modeling to support its threat scenarios. Some proposed sabotage methods are technically plausible but insufficiently detailed, leaving questions about real-world feasibility. The defensive measures are high-level and could be strengthened with concrete detection or mitigation strategies. Adding empirical case studies or simulations would make the work more practical and evidence-based.

**Questions:**

Can the authors provide quantitative or simulated case studies to estimate the real-world feasibility and impact of the proposed sabotage methods?

How do the identified attack vectors differ in difficulty and detectability across various AI training architectures (e.g., centralized vs. federated training)?

Could the authors expand on specific defensive mechanisms—technical or organizational—that could help detect or mitigate training sabotage in practice?

**Details Of Ethics Concerns:**

The paper discusses state-level AI sabotage techniques that could be dual-use if misapplied. An ethics review is recommended to ensure sensitive details are responsibly handled and publication aligns with ethical disclosure standards. Reviewers with expertise in AI security and dual-use governance should assess potential risks.

---

> ### Author Response · Authors · 2025-11-25
>
> We thank reviewer x7UV for their review, and hope that we have addressed their concerns. To make the work more concrete and immediately useful to security practitioners, we have created a checklist of informational questions they may wish to answer as part of their threat modelling assessment; this is attached as Appendix A.
>
> Due to dual-use concerns we have chosen not to publish a detailed feasibility assessment of specific attacks. Additionally, lack of public information on frontier training methods, lab security postures and adversary capabilities would substantially restrict the accuracy of any such analysis. However, we have added additional citations and commentary quantifying the feasibility of certain general classes of attacks we are concerned about, specifically corruption of pre-training data, gradual degradation via continuous learning, and sabotage of evaluations.
>
> The impact of different architectures on the efficacy of and mitigations for attack vectors was under-explored in our initial submission. We have added a substantial amount of commentary and illustration of this at the end of Section 3.3.
>
> For each of the identified attack vectors in Section 4, we have split out a paragraph about suggested mitigations and added more detail about their feasibility. Additionally, the attached practitioner checklist includes consideration of technical and organizational mitigations which may apply to most training runs in practice.

---

> ### Comment · Reviewer_x7UV · 2025-11-25
> **Looks better**
>
> Looks much better! Improved ratings.

---

### Official Review · Reviewer_XQsL · 2025-10-31

**Soundness:** 2
**Presentation:** 3
**Contribution:** 3
**Rating:** 6
**Confidence:** 4

**Summary:**

This paper investigates the possibility that nation states will sabotage frontier AI training runs of their rivals, causing the system to have degraded capabilities or hidden goals. Sabotage is seen as potentially less onerous than stealing model weights, which is an attack vector that has received a lot more attention. Sabotage might be less escalatory and therefore an attractive option for a nation state in a disadvantageous position compared to a more powerful rival. The paper presents some breakdown of the different types of sabotage to help understand the attack surface and attacker motivations.

**Strengths:**

A very interesting extension to MAIM and the RAND report on securing model weights. This paper fits more with what I think threat actors are likely to actually want to do in the real world. It is also clearly written from a position of expertise with how governments think and operate. The main text is clear and tells an interesting story.

**Weaknesses:**

The paper feels quite truncated now with some of the more sensitive parts removed. There aren't any experiments or even example applications of their taxonomy to different real world instances. I feel like I read a very detailed breakdown of a problem, but not a taxonomy.

It's worth noting that the citations in this work are nearly all in footnotes. I don't often see this in technical papers. I suppose it keeps the text more compact due to the citation style, but it is particularly unusual when a footnote has nothing except a citation in it. I think I would recommend merging some of the footnotes in. If the text gets too wordy, the paper can have more detailed sections that revisit topics. This is not a strong suggestion on my part, just an idea.

Specific comments on sections:

Section 2 is quite interesting and is probably new background for many readers. I think it could use more citations, especially in the second last paragraph. I had read that the Ukraine war had initially reduced the number of ransomware attacks on the rest of the world (though it's true that this trend seems to have reversed). This paragraph is also in the context of what's happening in wartime, which goes against the bullet point earlier “the situation does not yet justify overt military action”. Perhaps this list should use “or” rather than “and”. Or somehow reworded to include the Ukrainian war scenario.

The “salami slicing” done by China could use more citations as well, with the construction of artificial islands, Philippines, Vietnam, etc.

In section 3.1: The separation between the first stage (design and planning) and the second stage (data gathering and pre-processing) does not seem crisp. It seems likely that the second stage can proceed partially in parallel with the first stage. Also, creating a complete plan and deciding how large the model is going to be might depend fairly heavily on how much data can be obtained. The other stages are well known, pre-training and post training, and cannot really overlap. I wonder if it might make more sense to have a single first stage of planning, with several substeps that happen in parallel.

Typo in section 4: “efficacy of removal is a an open question.”

Minor typos in footnote 39: See is italic, and the semicolons aren't really grammatical.

**Questions:**

Why did you choose ICLR as a venue for this work? Overall, I am not sure whether it is the best venue. As someone in or adjacent to this field, I find it very interesting. Perhaps the machine learning community in ICLR could be one of the best audiences for this work, even if it is not a typical paper. But a dedicated AI safety conference/workshop may also be a good fit.

---

> ### Author Response · Authors · 2025-11-25
>
> We thank reviewer XQsL for the detailed review of our paper. We agree that the paper is somewhat truncated, given the removal of certain attack specifics, and so have used the taxonomy to create a checklist of general questions which practitioners may wish to answer as part of their threat modelling. These are divided according to the process timeline described in section 3.1 and are attached as Appendix A.
>
> We have merged in some of the footnotes which contained only a citation (and for which we did not feel it worsened the flow of the argument) and expanded others (and expect to merge more once a response is given to the final reviewer).
>
> We have added some additional sources relevant to Russia’s sabotage campaign in Europe, and clarified in a footnote that the current sabotage actions against countries other than Ukraine are precisely because Russia has proved unwilling to take direct military action against them (as opposed to ‘sabotage’ against Ukraine, which is more properly seen as military action instead) - i.e. we assert the original framing of ‘and’ is correct.
>
> We have added a footnote and three additional citations to provide readers more context on the “salami-slicing” in the South China Sea.
>
> Regarding section 3.1, we agree that stages overlap in practice. We present them sequentially for clarity in threat modelling, since our goal is to identify distinct attack surfaces (data sources vs. architectural decisions vs. training infrastructure) rather than describe project timelines. We've added a clarifying note in Section 3.1 as to why we present in this manner.
>
> Highlighted typographical issues have been addressed.
>
> We chose ICLR to reach the broad ML community who design and implement training systems. We strongly believe that effectively addressing this threat model will require awareness and buy-in from this community - the infrastructure engineers, data pipeline architects, and ML researchers whose work is critical to the creation of advanced AI systems and who (we believe) largely do not engage with specialized AI safety venues. We acknowledge that this is not a typical ICLR paper but assert that sabotage of training pipelines is under-considered in ML security research and that this framework provides actionable guidance for practitioners making design decisions.

---

### Official Review · Reviewer_Pskk · 2025-11-02

**Soundness:** 2
**Presentation:** 3
**Contribution:** 2
**Rating:** 4
**Confidence:** 4

**Summary:**

The paper argues that, for state actors, sabotaging AI training will often be more attractive than stealing model weights. It contributes (i) a threat-modeling framework covering strategic objectives (capability degradation vs. value misalignment; overt/covert; attributable/anonymous), (ii) a taxonomy of attack vectors (data poisoning, model poisoning, process disruption, with direct and indirect/organizational routes), and (iii) a walkthrough of the training pipeline highlighting plausible sabotage points and mitigations. It situates the analysis in historical context (e.g., Stuxnet; “grey-zone” operations) and discusses the MAIM (“Mutual Assured AI Malfunction”) dynamic. The Ethics Statement notes that some sensitive details were intentionally omitted; the authors also disclose substantial LLM assistance.

**Strengths:**

1. Makes a clear case that sabotage of training can be strategically favored in realistic scenarios (e.g., when compute deployment is infeasible for the attacker, or when preventing a rival’s capability matters more than copying it).

2. Useful decomposition by objective (degradation vs. misalignment) and overt/covert/attributable axes; ties choices to strategic implications and examples (Table 1).

3. Training pipeline stages are enumerated with concrete touchpoints for sabotage/mitigation—helpful for practitioners performing risk assessments.

4. The historical review (Stuxnet; Cold War precedents; grey-zone warfare) provides external validity; the paper clarifies where MAIM might or might not hold.

5. Dual-use considerations are acknowledged; sensitive technical details are intentionally withheld; LLM usage is disclosed.

**Weaknesses:**

1. Scope is largely qualitative, with limited technical novelty for ICLR. The piece reads more like a policy/security position paper than a machine-learning research paper; there are no models/algorithms, formal analyses, or empirical evaluations to test claims (e.g., attacker feasibility, detectability, or defense efficacy). Much of the content is synthesis + taxonomy. (The authors themselves note that a fuller threat-modeling study would require non-public information.)

2. The plausibility assessments (e.g., insider-driven model poisoning, RLHF data poisoning) are plausible but lack quantitative risk estimates (likelihoods, cost-to-attack, time-to-detect), red-team case studies, or operational measurements to anchor recommendations.

3. The paper cites J-AISI/NIST but would benefit from a side-by-side mapping showing exactly what is new (e.g., “process disruption” slice; organizational attacks) and what is reframed—ideally as a comparison table or figure.

4. As a primarily strategic/cybersecurity paper with no ML experiments, it may struggle to meet ICLR’s contribution bar without a stronger technical component (e.g., detection methods, formal threat models with testable implications, or empirical audits across pipelines). (ICLR asks that reviews focus on value/new knowledge; here the value is practical but the ML research delta is thin.)

**Questions:**

1. Re pperationalization: Can you include at least one worked case study (anonymized) that quantifies attacker cost, access required, expected detection latency, and impact, for one data-poisoning and one process-disruption path? This would materially strengthen practitioner utility.

2. Re comparative taxonomy: Provide a comparison matrix vs. J-AISI/NIST categories to make the paper’s unique framing absolutely explicit (what’s added vs. relabeled).

3. Re MAIM implications: Could you formalize simple game-theoretic conditions (signals, attribution noise, cost curves) under which sabotage dominates theft, and simulate parameter regimes to test robustness of your MAIM-related claims?

4. Re Defense playbooks: Your mitigations are high-level; can you attach a practitioner checklist per pipeline stage (e.g., specific logging/audit artifacts for RLHF datasets, insider-risk controls, compromise-resilience drills)?

5. Re Responsible disclosure: Since some analyses were redacted, can you clarify what can be shared with vetted reviewers under confidentiality to better evaluate your novelty/insight claims?

**Details Of Ethics Concerns:**

No ethical violations evident; dual-use handled responsibly. Paper explicitly discloses use of Claude Sonnet 4 for drafting, no issues from me.

---

> ### Author Response · Authors · 2025-11-28
>
> We thank reviewer Pskk for their review, and are confident that in responding to it we have substantially strengthened the actionable utility of this work.
>
> With regards to specific questions:
>
> 1. Due to dual-use concerns we have decided not to do this in our public output (see the response to reviewer x7UV). However, part of our response to Q4 includes prompts to assist practitioners in considering these factors since we believe it is more usefully done in situ given the relevance of non-public knowledge.
>
> 2. We have created a comparative taxonomy for classification of threats between this paper and the J-AISI/NIST work; this is attached as Appendix B - we hope this is what you were looking for?
>
> 3. We found this suggestion interesting but believe it would be too far outside the current scope of our threat modelling contribution, such analysis not being typical for this type of paper (cf. RAND's model weights report). However, we strongly believe that this is useful and necessary work for proper understanding of the MAIM dynamic and have added commentary to this effect in a footnote to Section 5.
>
> 4. We have created a checklist which provides practitioners a set of questions to consider for each pipeline stage, including the practicality of different mitigations; this is attached as Appendix A.
>
> 5. We have received, and agreed, a specific request to share an unredacted version with relevant figures within [specific governmental body redacted pending review by Senior ACs]. That version will be largely the same as this but with additional details on the feasibility of the attacks listed, as well as similar treatment of additional threat vectors we identified during our threat modelling* but did not include in this version because we consider them sufficiently novel and unmitigated, easily exploitable, or currently difficult to mitigate (due to a mixture of technical, organisational, and political factors), that it would be irresponsible to publicise their existence at this time. We will continue to engage in discussion with them about this work and its implications, and we will follow their guidance on any eventual distribution of this material.
>
> *In the specific, singular case of the “adversaries attempt to sabotage evaluations so that labs do not detect other sabotage, or release unintentionally dangerous models” threat - we do believe this to be novel to the public domain but a) almost certainly already considered by state actors and b) it is sufficiently important that practitioners consider this when designing and reviewing evals as to justify our drawing attention to it.

---

### Meta-Review · Area_Chair_583u · 2026-01-11

**Summary:**

The paper presents a qualitative threat modeling framework for the sabotage of frontier AI training runs by state actors. While the reviewers acknowledged the importance and originality of the topic, specifically the shift in focus from weight theft to capability degradation, the core concern is the paper's poor fit for ICLR.

The primary reasons for rejection are a lack of technical novelty, the absence of empirical or quantitative analysis, and a reliance on policy-oriented synthesis over machine learning research. Despite revisions, the work remains a high-level security position paper that does not provide the testable implications or algorithmic contributions expected at a top-tier technical conference.

**Reviewer Concerns:**

Addressed by Rebuttal
* Lack of Practitioner Guidance: The authors added Appendix A (a checklist) and Appendix B (a comparison matrix vs. J-AISI/NIST) to provide more "actionable" content.
* Contextual Clarity: Minor issues regarding citation style and historical context for "grey-zone" operations were corrected to satisfy Reviewer XQsL.
* Architectural Differences: The authors added commentary on how sabotage differs across training architectures.

⠀Outstanding Concerns (Reasons for Rejection)
* Technical "Delta" and Venue Fit: Reviewer Pskk and Reviewer x7UV both noted the lack of models, algorithms, or formal analyses. The paper does not offer new ML knowledge, but rather a synthesis of existing security concepts applied to ML.
* Lack of Empirical Validation: Reviewers repeatedly asked for quantitative risk estimates, cost-to-attack simulations, or red-team case studies. The authors declined to provide these, citing dual-use concerns and a lack of public information.
* Truncated Content: Reviewer XQsL noted the paper feels "truncated" due to the removal of sensitive details. This lack of transparency, while perhaps necessary for security, prevents the scientific community from properly evaluating the novelty of the specific threat vectors mentioned.

**Reviewer Scores:**

- Reviewer XQsL: 6->4
Although the reviewer was initially positive, they expressed significant doubt about the venue fit and noted the paper was "truncated". Without the inclusion of experiments or real-world applications, the "fair" soundness rating remains a barrier.
- Reviewer x7UV: 4->4
Despite the "looks better" comment, the core weakness they identified, i.e., a lack of empirical validation and detailed technical feasibility, remains unaddressed in the final version due to the authors' redaction policy.
- Reviewer Pskk:  4->4
While they appreciated the new appendices, their fundamental critique was the "thin" ML research delta and lack of quantitative anchor points. The rebuttal did not add the technical components requested.

---

### Decision · Program_Chairs · 2026-01-26

Reject